# Inhibitory Effects of Bovine Lactoferricin-Lactoferrampin on Senecavirus A and Foot-and-Mouth Disease Virus with Recombinant Lactobacillus Oral Treatment in Mice

**DOI:** 10.3390/vetsci12030199

**Published:** 2025-02-25

**Authors:** Wenyue Zhao, Senhao Zhang, Ling Sui, Xiaona Wang, Jiaxuan Li, Wen Cui, Yanping Jiang, Xinyuan Qiao, Lijie Tang

**Affiliations:** 1College of Veterinary Medicine, Northeast Agricultural University, Harbin 150030, China; zwy1514@outlook.com (W.Z.); zhang_sen_hao@163.com (S.Z.); isseven111@163.com (L.S.); xiaonawang0319@163.com (X.W.); lijiaxuan.1993@163.com (J.L.); cuiwen_200@163.com (W.C.); jiangyanping@neau.edu.cn (Y.J.); 2Heilongjiang Key Laboratory for Animal Disease Control and Pharmaceutical Development, Harbin 150030, China

**Keywords:** foot-and-mouth disease virus, Senecavirus A, bovine lactoferricin-lactoferrampin, antioxidant, recombinant *Lactobacillus reuteri*

## Abstract

Foot-and-mouth disease virus (FMDV) and Senecavirus A (SVA) can cause vesicular disease in pigs, causing serious harm to the livestock industry. There is, therefore, an urgent need to develop antiviral drugs for prevention and/or treatment. In this study, we investigated the antiviral activity of bovine lactoferrin peptide. We found that bovine lactoferrin peptide plays an antiviral role in cases of infection with FMDV and SVA, as well as an anti-oxidative role, reducing virus-induced oxidative stress. The oral administration of LFCA-expressing recombinant lactic acid bacteria protected mice from SVA injury.

## 1. Introduction

Foot and mouth disease (FMD) is a highly contagious zoonotic disease caused by the foot-and-mouth disease virus (FMDV) in cloven-hoofed animals [1,2]. Similarly, Senecavirus A (SVA), a member of the Picornaviridae family like FMDV, causes vesicular disease in pigs and was first discovered in 2002 during cell culture experiments [3,4,5,6]. Both viruses are challenging to control and both pose significant risks to the global livestock industry due to their widespread transmission [7,8,9,10].

Lactoferrin (LF) is produced by breast cells and white blood cells in tissues and is involved in blood coagulation regulation and NK cell activation, as well as the production of interleukin and tumor necrosis factor. LF is found in the secretions of most mammals; such secretions include milk, tears, and saliva [11,12,13]. Its derivative peptides possess antibacterial [14,15,16], antiviral [17,18], antifungal [19], anti-inflammatory [20], and anticancer [19,21,22,23] properties [24,25,26]. Recent studies have indicated that bovine lactoferrin peptides can promote osteoblast differentiation [27,28]. The peptides lactoferricin (Lfcin) and Lactoferrampin (Lfampin) have been extensively studied, and a chimeric structure containing both has been designed to enhance their antimicrobial activity [29,30,31]. Other studies have suggested that the antiviral effects of lactoferrin peptides involve competition with viruses for common cell membrane receptors [32,33]. Lfcin, in particular, is believed to inhibit viral entry through interactions with molecules such as heparan sulfate and glycosaminoglycans on the cell surface [34,35]. Studies on herpes simplex virus (HSV) have shown that Lfcin may also exhibit intracellular antiviral activity by inhibiting viral synthesis within the cell [36,37,38]. Most recent studies on Lfampin have focused on its antibacterial effect. In contrast, there have been few studies on its antiviral effect, and further research is still needed in this regard; however, the structure of Lfampin suggests that it may exhibit antiviral properties.

In this study, we investigated the antiviral activity of the chimeric peptide lactoferricin-lactoferrampin (LFCA). Our results demonstrated that LFCA strongly inhibits the replication of both FMDV and SVA. Additionally, LFCA provides protection to cells prior to viral infection by enhancing the transcription of antioxidant genes and reducing oxidative stress damage. The recombinant lactic acid bacteria expressing LFCA were constructed, and the oral characteristics of the bacteria were used for in vivo experiments, which further proved that LFCA has a protective effect on SVA infection in mice.

## 2. Materials and Methods

### 2.1. Cells and Viruses

BHK-21 cells and SVA (LJ-2021) strains were provided by the Microbiology and Immunology Laboratory. PK-15 cells and FMDV strains were provided by the Lanzhou Veterinary Research Institute of the Chinese Academy of Agricultural Sciences. LFCA synthetic products were provided by Comate Bioscience Co., Ltd., located in Changchun City, Jilin Province, China.

### 2.2. Culture Conditions for BHK-21 and PK-15 Cells

The frozen cells were removed from a liquid nitrogen tank and quickly thawed in a 37 °C water bath. The cells were transferred to a centrifuge tube and re-suspended in a preheated DMEM complete medium (37 °C). The sample was centrifuged at 1000 rpm for 5 min; then, the cells were gently resuspended in 5 mL DMEM complete medium supplemented with 10% fetal bovine serum (FBS). Next, the suspended cells were transferred to 25 cm^2^ cell culture bottles. The flask was placed in a 37 °C incubator with 5% CO_2_. After 6 h, the medium was replaced with DMEM and the cells continued to be cultured. When the cells covered about 90% of the bottom of the culture bottle, the passage culture was continued. The medium was discarded, the cells were washed 3 times with PBS, and 1 mL of cell dissociation solution (0.25% trypsin + 0.02% EDTA) was added. When the cells were separated from the bottom of the culture bottle, an appropriate amount of DMEM was added to completely stop the digestion of the culture medium and gently absorb the culture bottle wall to form a cell suspension.

### 2.3. Cell Activity Assay

BHK-21 and PK-15 cells were seeded at a density of 1 × 10^5^ cells/mL in a 96-well plate until 90% confluence was reached. Next, synthetic lactoferricin-lactoferrampin (LFCA), beginning at an initial concentration of 1 mg/mL, underwent serial dilutions using serum-free DMEM to yield concentrations of 400 µg/mL, 200 µg/mL, 100 µg/mL, 50 µg/mL, 25 µg/mL, 12.5 µg/mL, and, finally, 6.25 µg/mL. After discarding the medium from the wells, 100 µL of each concentration was dispensed into the wells and subsequently incubated at 37 °C for 4 h. Following incubation, CCK-8 reagent was diluted in DMEM at a ratio of 1:100 according to the manufacturer’s instructions, with 100 µL of the diluted reagent added to each well. The plate was then incubated for 1 h at 37 °C, being protected from light, and the OD_450 nm_ absorbance was measured using a microplate reader. Cell viability (%) was calculated using the following formula: cell viability (%) = [A(treated) − A(blank)/A(control) − A(blank)] × 100.

### 2.4. Determination of Viral Copy Number by RT-qPCR

BHK-21 and PK-15 cells were seeded at a density of 1 × 10^5^ cells/mL into 24-well plates. The cells were cultured at 37 °C with 5% CO_2_ until they reached approximately 80% confluence. LFCA was diluted with DMEM to a concentration of 1 mg/mL. The experiment was divided into five groups, each with three parallel wells. SVA was applied at an MOI (multiplicity of infection) of 0.01, and FMDV at an MOI of 0.001. The experimental groups were as follows: (a) Pretreatment Group: The cells were treated with LFCA at a concentration of 150 µg/mL for 4 h. Subsequently, SVA and FMDV were inoculated onto BHK-21 and PK-15 cells, respectively, at multiplicities of infection (MOIs) of 0.01 and 0.001. Following an hour-long incubation period, the cells were washed thoroughly three times with phosphate-buffered saline (PBS), and the medium was then replaced. (b) Adsorption Phase Group: LFCA (150 µg/mL) was mixed with SVA and FMDV at MOIs of 0.01 and 0.001, respectively. The mixtures were then inoculated onto BHK-21 and PK-15 cells. After 1 h of incubation at 37 °C, the medium was replaced. (c) Entry Phase Group: SVA and FMDV were inoculated onto BHK-21 and PK-15 cells at MOIs of 0.01 and 0.001, respectively. After an hour of incubation, LFCA (150 µg/mL) was added to the medium. (d) Replication Phase Group: SVA and FMDV were inoculated onto BHK-21 and PK-15 cells at MOIs of 0.01 and 0.001, respectively. The medium was changed after it had been incubated for 1 h, and LFCA (150 μg/mL) was added to the medium after 2 h had passed. (e) Infection Control Group: The cells were washed three times with PBS. Subsequently, SVA and FMDV were inoculated onto BHK-21 and PK-15 cells at MOIs of 0.01 and 0.001, respectively. After an hour of incubation, the medium was replaced. Virus supernatants for SVA and FMDV were collected at 12 and 24 h post-infection, respectively. The viral RNA was extracted from each sample and underwent reverse transcription. RT-qPCR was then conducted to determine the viral copy number for each virus.

### 2.5. The Half-Maximal Inhibitory Concentration of LFCA Against the Replication of SVA and FMDV

The cells were treated with LFCA at final concentrations ranging from 400 μg/mL to 0 μg/mL for 4 h. Following treatment, the cells were washed three times using a PBS solution. Subsequently, BHK-21 and PK-15 cells were inoculated with SVA and FMDV, respectively, at MOIs of 0.01 and 0.001. The cells were then incubated at 37 °C for 1 h before the medium was replaced. Viral RNA was extracted at 12 h and 24 h post-inoculation and used as a template for quantifying the viral copy number through RT-qPCR.

### 2.6. Detection of Cell Oxidative Stress Index

To detect ROS levels, the reactive oxygen species (ROS) kit was operated according to the instructions. The cells were treated with ultrasound at 30 W power, at intervals of 3–5 s every 30 s, repeated 4 times. The levels of super oxide dismutase (SOD) activity, Glutathione peroxidase (GSH-PX) activities, and malondialdehyde (MDA) production in the treated samples were assessed according to the instructions provided with the kit.

### 2.7. The Alterations in the Cellular Antioxidant Gene Expression Levels

The cells were pretreated with LFCA, and cell samples were collected at 6 h and 12 h after SVA inoculation, as well as at 4 h and 6 h after FMDV inoculation. The mRNA levels of nuclear factor erythroid-derived 2-like 2 (*Nrf2*), heme oxygenase-1 (*Ho-1*), and NAD(P)H dehydrogenase (quinone) 1 (*Nqo1*) were analyzed using RT-qPCR, while Western blot was utilized to detect the protein level. β-actin expression served as the internal reference for quantitative data analysis.

### 2.8. Detection of Antiviral Activity of pPG-A3-LFCA/LRco21 In Vitro

A recombinant *Lactobacillus reuteri* that can secrete and express LFCA was previously constructed and given the name pPG-LFCA/LRco21. The A3 signal peptide gene fragment from *L. reuteri* was linked to the expression vector pPG-LFCA through double enzyme digestion and given the name pPG-A3-LFCA. The conjugated product was transferred into the receptive state of Escherichia coli TG1 for cloning, and the extracted plasmid was electrotransformed into the receptive state of *Lactobacillus reuteri* LRco21. The sequence of A3 signal peptide is 5′-ATGGTGAACCGAGATAAATTTCGTTTTTCGACTGCTTAAGCGGATATTTATAATAGGACTTTTAGTAGGTGGAGGATGGCTTTATTTCAATGATGCACAAGTTCAAGCAACGGCG-3′, and the sequence of LFCA is 5′-TTTAAATGCCGTCGTTGGCAATGGCGCATGAAGAAGTTGGGCGCTCCGAGTATTACCTGCGTTCGCCGCGCTTTTGGCGGTGGCTCCAGTGTTGATGGCAAAGAAGATTTGATTTGGAAATTGTTGAGTAAAGCTCAAGAAAAATTTGGCAAAAATAAAAGTCGTTAA-3′. The recombinant strains pPG-LFCA/LRco21, pPG-A3-LFCA/LRco21, and LRcon were incubated at 37 °C for 16–20 h and centrifuged at 4 °C, 5000 rpm for 10 min. Culture supernatants were collected and concentrated using trichloroacetic acid precipitation (TCA) and quantified using indirect competitive ELISA. The antibody used in ELISA and Western blot was the myc-tagged antibody on the carrier. The bacterial precipitate was broken using an ultrasonic instrument. The cells were treated with the bacterial precipitation after filtration for 4 h. Subsequently, SVA and FMDV were inoculated onto BHK-21 cells and PK-15 cells, which were collected at 12 h and 4 h, respectively, and the virus copy number was detected using RT-qPCR.

### 2.9. Protective Effect of LFCA on Virus Infection in Animals

Thirty ICR suckling mice aged 1 to 4 days were selected and divided into three groups: control group (CON), infection group (SVA), and administration group (LFCA). On the first day of the experiment, each suckling mouse in the control group and the infected group was given 20 µL PBS, and each mouse in the administration group was given recombinant *Lactobacillus reuteri* pPG-A3-LFCA/LRco21 (1 × 10^8^ CFU). On the fourth day, each mouse in the infected group and the administration group was given 20 µL SVA (3.75 × 10^6^ PFU/mL); also on the fourth day, each mouse in the control group and the infected group was given 20 µL PBS. During this period, the symptoms and manifestations of SVA infection were observed. The mice were then euthanized by cervical dislocation on the fifth day after SVA infection.

### 2.10. Statistical Analyses

The statistical software IBM SPSS Statistics 26 was used for one-way analysis of variance (one-way ANOVA), and Origin 2021 was used for the statistical analysis of the experimental data. All experiments were conducted with three independent replicates (*n* = 3). * represented significant difference (*p* < 0.05); ** represented extremely significant difference (*p* < 0.01); # represented significant difference (*p* < 0.05); and ## represented extremely significant difference (*p* < 0.01).

## 3. Results

### 3.1. Cytotoxicity Assessment of LFCA

After treatment with LFCA of different dilutions, the cell survival rate was about 100% (Figure 1), and LFCA at concentrations of less than 400 μg/mL had no toxicity to cells.

### 3.2. LFCA Affected SVA and FMDV Replication

The data show that bovine lactoferrin peptides could inhibit the replication of SVA and FMDV at all stages (Figure 2). The inhibition rate could reach 94.9% after pretreatment before SVA infection. The inhibition rates of the virus in the stages of adsorption, invasion, and replication were 91.2%, 75.6%, and 65.9%, respectively, all of which were lower than that of the Pretreatment Group. The highest inhibition rate of FMDV replication was observed in LFCA-treated cells (74.3%). The inhibition effect of the adsorption stage was second only to that of the replication stage, and the inhibition rate was 63.4%.

### 3.3. LFCA Inhibited Virus Replication at 50% Inhibitory Concentration

The inhibitory effects of LFCA on the replication of both SVA and FMDV were confirmed by calculating the inhibitory rates at various LFCA concentrations (Figure 3). The inhibition rates of SVA by LFCA were 41.8%, 57.3%, 80.3%, 89.6%, 94.5%, and 96.1%. The 50% inhibitory concentration of LFCA against SVA was determined to be 8.96 μg/mL. LFCA demonstrated an inhibition rate of 66.3% against FMDV at 400 μg/mL; this rate decreased with lower concentrations and was absent at 50 μg/mL. Data analysis revealed that the 50% inhibitory concentration (IC_50_) of LFCA against FMDV was 203.9 μg/mL.

### 3.4. Effect of LFCA on Oxidative Stress Induced by Virus Infection

At 6 h, the copy number of SVA in the LFCA group was significantly lower than that in the SVA group (*p* < 0.01), and LFCA inhibited the replication of SVA (Figure 4a). LFCA could significantly reduce the intracellular ROS level of SVA- and FMDV-infected cells and alleviate the oxidative stress response of cells (Figure 4b). GSH-PX activity (Figure 4d) and the content of MAD (Figure 4e) were significantly decreased in the LFCA-treated group (*p* < 0.01) due to the decreased oxidative stress level. However, there was no significant difference in SOD activity between the LFCA-treated group and the virus-infected group (Figure 4c).

### 3.5. Effect of LFCA on Expression of Antioxidant Genes

LFCA treatment significantly increased the mRNA levels of Nrf2, HO-1, and NQO1 genes in cells infected with SVA and FMDV. This was revealed by quantitative data anal-ysis (Figure 5a). At 12 h, the mRNA levels of Nrf2, Ho-1, and Nqo1 genes in LFCA-treated cells were not significantly different from those in SVA-infected cells. Similarly, at 6 h, there was no significant difference between the mRNA levels of these genes in LFCA-treated and FMDV-infected cells. The protein expression levels of Nrf2, Ho-1, and Nqo1 genes in SVA-infected cells at 6 h and 4 h after infection were significantly increased (Figure 5b). However, the expression levels of Nrf2 and HO-1 in SVA-infected cells were significantly lower than those in SVA-infected cells after LFCA treatment, and there was no significant difference between the expression of NQO1 and SVA-infected cells. After treatment with LFCA, the expression of Ho-1 in cells infected with FMDV was significant-ly lower than that in cells infected with FMDV, while the expression of Nrf2 in cells in-fected with FMDV was not significantly different from that in cells infected with FMDV. Only the expression of NQO1 in cells infected with FMDV was significantly higher than in those infected with FMDV.

### 3.6. Identification of LFCA Expression in Recombinant Lactobacillus

The signal peptide of the recombinant strain was optimized, and its expression was verified (Figure 6a,b). The secreted LFCA was quantified using ELISA, and the expression level was found to be as high as 1.12 μg/mL (Figure 6c), which was higher than that of the original recombinant strain preserved in the laboratory. pPG-A3-LFCA/LRco21 was used in subsequent experiments. The expression of LFCA in the constructed recombinant lactic acid bacteria pPG-A3-LFCA/LRco21 was detected using confocal laser microscopy (Figure 6a) and Western blot (Figure 6b-ii), and its inhibitory effect on the replication of SVA and FMDV in vitro was verified; this proved that the LFCA expressed by the recombinant bacteria had antiviral activity (Figure 6d).

### 3.7. The Recombinant Lactobacillus Showed a Protective Effect Against SVA Infection in Mice

The observed symptoms in ICR mice infected with SVA at four days of age included tremors, hind limb paralysis, slow movement, and inability to stand. Compared with the administration group, symptoms in the infected group appeared sooner and were more pronounced (Figure 7a and Table 1). After five days of infection, symptoms in both the SVA-infected and administration groups showed some degree of improvement (Figure 7c). SVA was detectable in the heart, liver, spleen, lungs, kidneys, brain, and intestines in both infected and treated mice. Notably, the viral load was highest in the lungs. In the lungs and intestines, the viral loads in the treated group were significantly lower than those in the infected group (*p* < 0.01). In other tissues, such as the heart, spleen, and brain, the viral load was lower; however, a significant difference was still observed between the treatment and infection groups (*p* < 0.05). The lungs with high viral load were selected for HE staining (Figure 7b). The morphology of the lung tissues in the control group was normal, but the lung tissues in the infected group showed obvious thickening of the alveolar wall, as well as lung parenchyma, alveolar collapse, and mild inflammatory infiltration. The pathological changes in the lung tissue of the mice in the administration group were significantly alleviated, compared with those in the infection group, with only slight thickening of the alveolar wall being exhibited, and no obvious inflammatory infiltration. These results showed that the recombinant strain expressing LFCA could protect mice from SVA infection.

## 4. Discussion

The results of this study demonstrate the significant antiviral potential of bovine lactoferricin-lactoferrampin (LFCA) against both foot-and-mouth disease virus (FMDV) and Senecavirus A (SVA). By targeting these viruses, LFCA could become a valuable tool in managing infectious diseases that pose a substantial threat to the livestock industry.

Bovine lactoferrin peptide has many biological functions, and the mechanisms of these functions have been studied. The function of bovine lactoferrin peptide which has been most widely studied and applied is its antibacterial activity [39]. Some studies have shown that bovine lactoferrin peptide can penetrate the cell membrane and exhibit intracellular activity, inhibiting bacterial growth; it has also been shown to kill Candida albicans by disrupting cell membranes [34,40], inducing reactive oxygen species (ROS) production, and causing mitochondrial dysfunction [41]. Because there have been few studies on the effect of bovine lactoferrin peptide on porcine microRNA virus, in the present work, we explored the anti-FMDV and anti-SVA effects of LFCA. Our results showed that LFCA could inhibit the replication of SVA and FMDV in four stages, with a high inhibition rate in the early stage. These high inhibition rates highlight the potent antiviral properties of LFCA, and it is suggested that bovine lactoferrin peptide may play an important role in the early period of the virus replication cycle. This is consistent with the conclusion that bovine lactoferrin peptide can competitively bind to cell surface receptors [34,35]. At the same time, LFCA can play a certain role in each stage of SVA and FMDV replication, according to the conclusion of other researchers that LFcin can play an anti-HIV role in cells [38], suggesting that LFCA may play a role in cells. The potential mechanisms by which LFCA exerts its antiviral effects include competition with viruses for cell membrane receptors and the intracellular inhibition of viral synthesis. These two mechanisms contribute to its efficacy in reducing viral loads and protecting cells. Due to the antioxidant effect of LFCA itself, we also attempted in the present study to explore the role of LFCA in the intracellular oxidative stress response caused by SVA and FMDV infection. Our results showed that virus infection could stimulate oxidative stress in cells, and the level of oxidative stress in cells decreased significantly after LFCA treatment. Previous research used in vivo models to evaluate the effect of Lfcin on oxidative stress and found a significant reduction in oxidative stress [42], which is consistent with the conclusion of our research in vitro. The significant increase in SOD activity suggests that LFCA may play a role in the antioxidant process.

In terms of antioxidant gene expression, our findings revealed that LFCA can also enhance the transcription of antioxidant genes in the early stage of virus infection, exert an antioxidant effect, and reduce cell damage caused by oxidative stress. Our results are consistent with the conclusion proposed by previous researchers that SARS-CoV-2 infection significantly suppressed NRF2 antioxidant response, generating oxidative stress conditions to its advantage [43]. Therefore, we hypothesize that LFCA may contribute to its antiviral mechanisms by modulating the antioxidant pathway. Further investigations will be undertaken in future research to explore this possibility. At the later stage, the expression level of antioxidant genes was similar to that of the virus-infected group due to cell compensation. After LFCA treatment, the trend in protein expression was not completely consistent with the transcription level, with only a part of antioxidant gene protein expression being increased. It is presumed that this phenomenon was caused by the different translation stages of different proteins and the relative lag in protein expression. NQO1 may be more involved in the antioxidant activity of LFCA, compared with other antioxidant genes, and further study is warranted in this regard. Our results also showed that the antiviral effects of LFCA on SVA and FMDV at different stages of replication were different. One of the reasons for this may be that the replication rate of FMDV is so fast that the antiviral effect of LFCA in the early stage is weakened, and the effect of LFCA in the later stage is not obvious. From the overall rates of inhibition of LFCA-treated cells against the two viruses, it was determined that the blocking ability of LFCA against FMDV was significantly lower than that of SVA. This may be related to the invasion and replication mechanisms of the two viruses; here, again, further research is needed.

In in vivo experiments, viral loads in mouse tissues after 5 days of SVA infection were found to be generally low, because lab-preserved SVA strains did not have a strong ability to infect ICR mice. Previous studies on the pathogenicity of SVA in mice and piglets have shown high pathogenicity, which is different from the results obtained in our research [44,45]. After analysis, this difference is due to the virulence of the strain and the culture method of the strain. The use of primitive animals for passage is more conducive to the recovery of virulence of the strain, and we will learn from the experience for further research in the future. Although the viral loads in lung and intestinal tissues were low, the differences in viral loads indicated that the recombinant strain expressing LFCA had a significant protective effect against SVA infection, and the pathological changes in lung tissues of the experimental group of mice were significantly alleviated after the oral administration of the recombinant strain. The reduction in viral load was more significant in the gut than in the lung, indicating that the application of LFCA in animals has great potential. These findings provide data support for the future clinical application of LFCA and validate the potential of LFCA as a preventive and therapeutic agent.

## 5. Conclusions

This study underscores the promising antiviral potential of bovine lactoferricin-lactoferrampin (LFCA) against FMDV and SVA. By inhibiting viral replication, reducing oxidative stress, and enhancing antioxidant gene expression, LFCA demonstrates both immediate and long-term protective effects. The successful use of recombinant *Lactobacillus reuteri* expressing LFCA in mice further validates its efficacy as an antiviral agent. Given the significant impact of FMDV and SVA on the livestock industry, LFCA could be developed into an effective antiviral drug for the prevention and treatment of these diseases. Future research should focus on optimizing the dosage, delivery methods, and exploring LFCA’s effects on other viral pathogens.

These findings offer a promising avenue for combating viral infections in livestock, with the potential to safeguard animal health and improve industry resilience.

## Figures and Tables

**Figure 1 vetsci-12-00199-f001:**
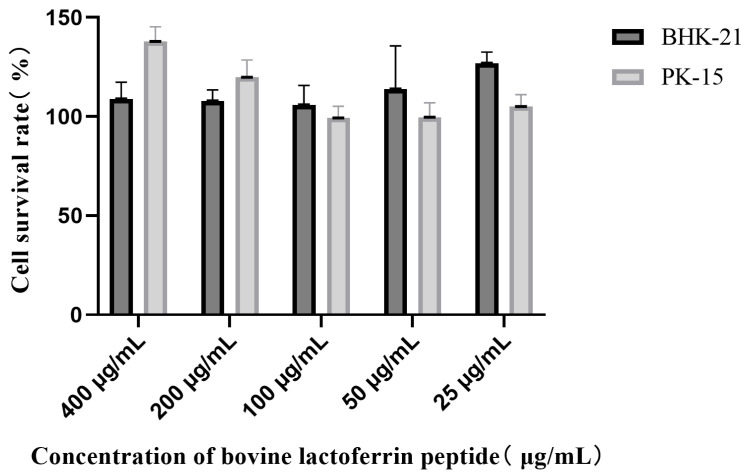
Effect of LFCA on the survival of BHK-21 cells and PK-15 cells. The two colored columns represent the survival rates of BHK-21 cells and PK-15 cells.

**Figure 2 vetsci-12-00199-f002:**
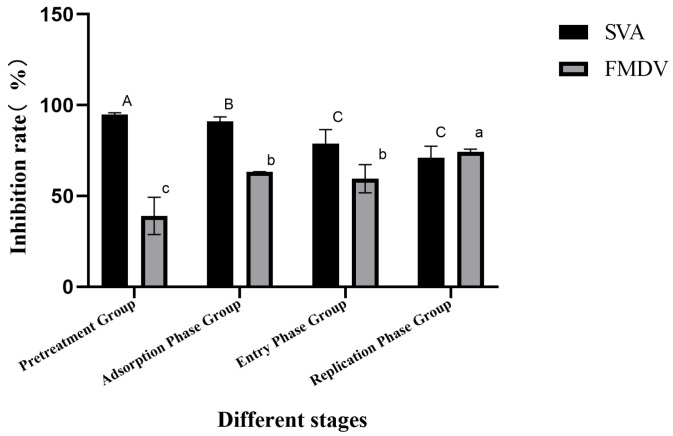
Effects of LFCA on SVA and FMDV replication. A, B, C and a, b, c represent the differences between the various stages of SVA and FMDV, respectively.

**Figure 3 vetsci-12-00199-f003:**
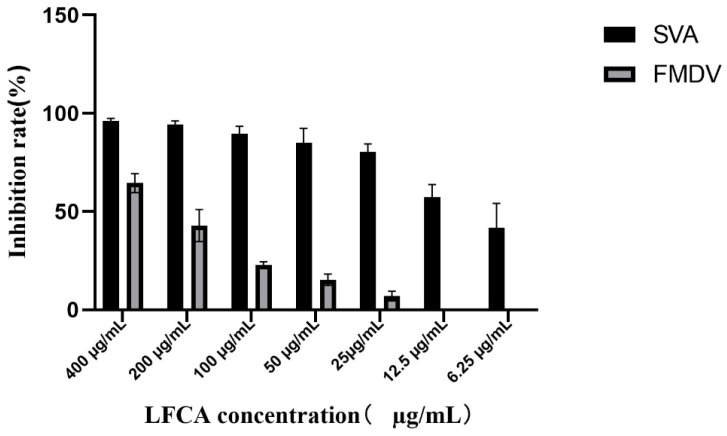
The inhibition rates of SVA and FMDV by different concentrations of LFCA.

**Figure 4 vetsci-12-00199-f004:**
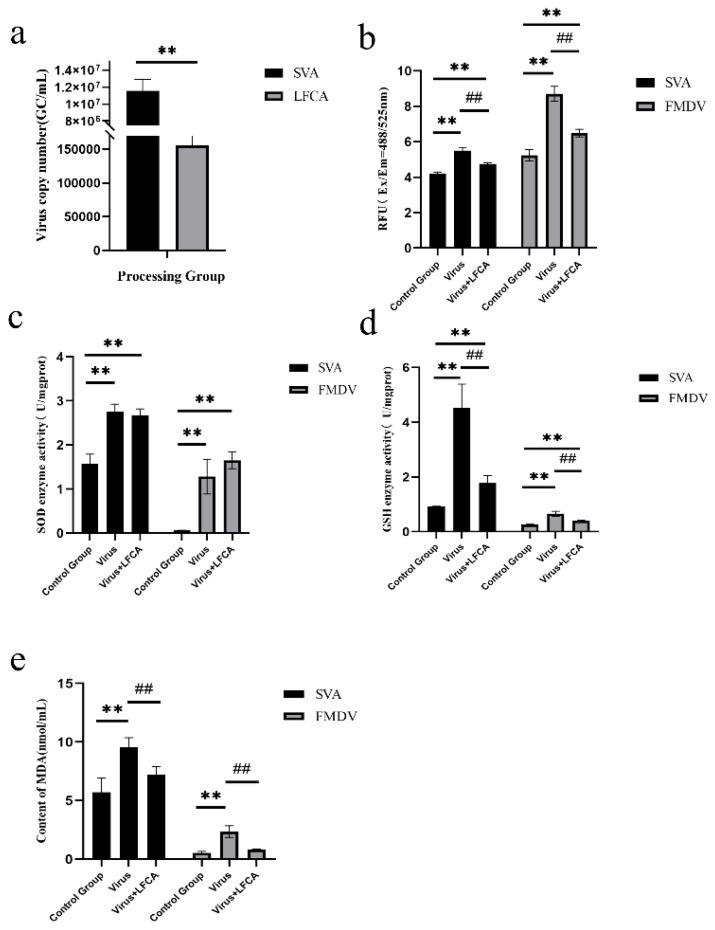
Effects of LFCA on oxidative stress induced by SVA and FMDV infection. (**a**) Viral copy number of LFCA-treated cells inoculated with SVA for 6 h. (**b**) Changes in intracellular ROS levels in cells (**c**) Changes in SOD activity in cells. (**d**) Changes in GSH activity in cells. (**e**) Changes in MDA levels in SVA-infected cells. ** *p* < 0.01; ## *p* < 0.01; * Significance of differences between representatives and controls. # Significance of the difference between the representative and virus inoculation group.

**Figure 5 vetsci-12-00199-f005:**
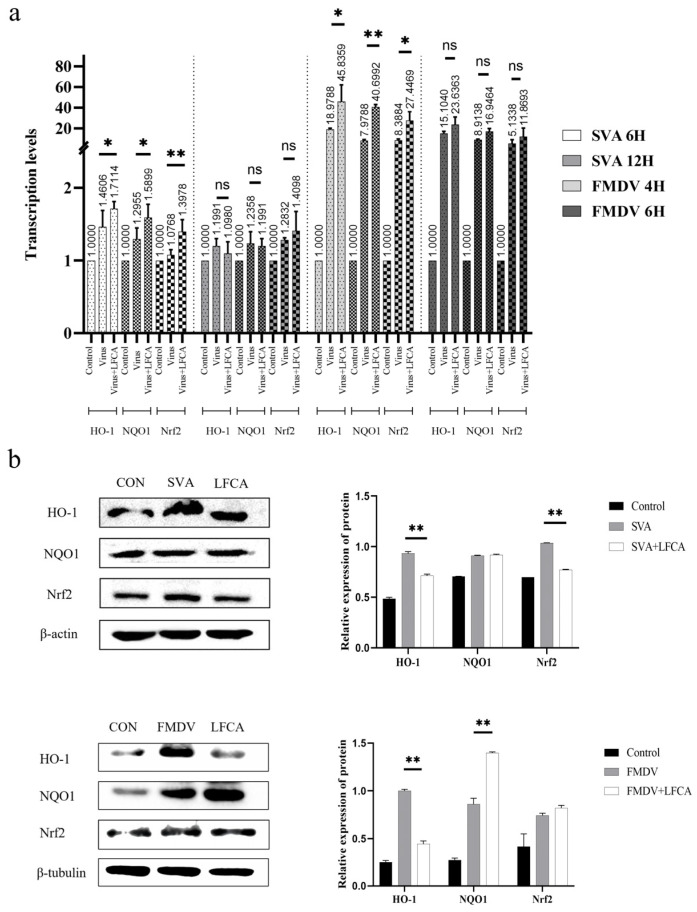
Effects of LFCA on the expression of intracellular antioxidant factors. (**a**) mRNA levels of anti-oxidative genes were detected at 6 h and 12 h after infection with SVA, and at 4 h and 6 h after infection with FMDV. (**b**) Protein levels of anti-oxidative genes in the SVA group at 6 h and the FMDV group at 4 h after infection, using gray-scale analysis to quantitatively analyze the protein (see Appendix A). ** *p* < 0.01; * *p* < 0.05; * significance of the difference between the representative and virus inoculation group.

**Figure 6 vetsci-12-00199-f006:**
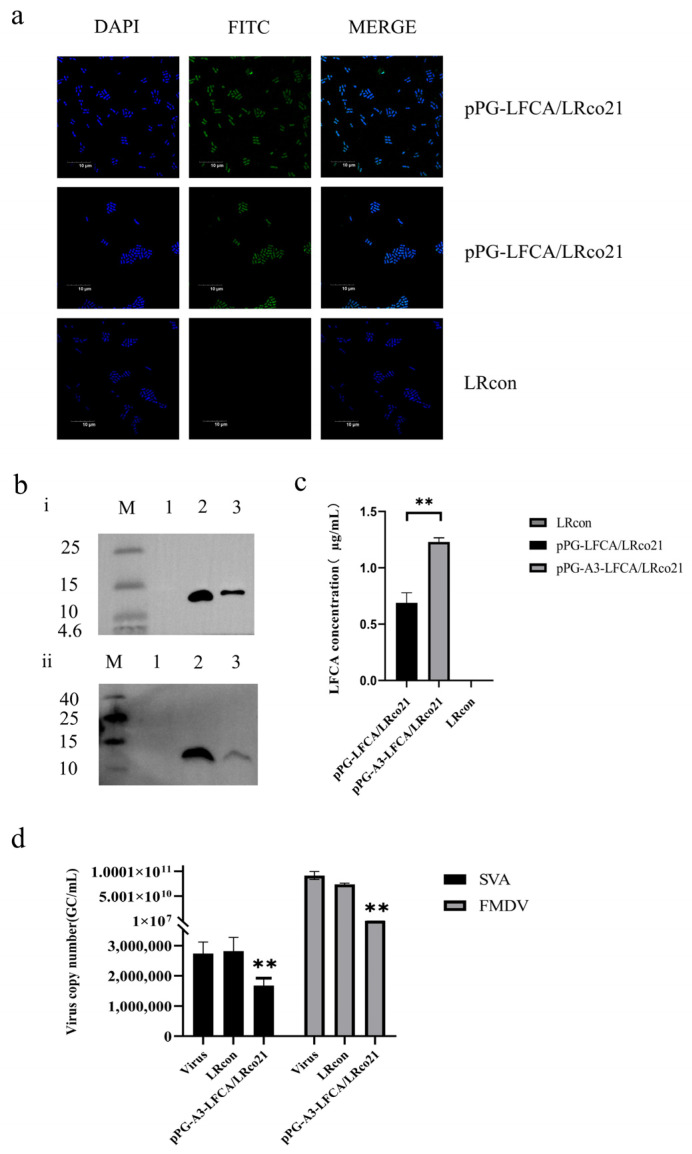
Identification of LFCA expression. (**a**) Laser confocal microscopy was used to detect the expression of LFCA in bacteria. (**b**) i. Western blot verified the expression of LFCA in LRcon (Lane 1), pPG-A3-LFCA/LRco21 (lane 2), and pPG-LFCA/LRco21 (Lane 3); ii. Western blot was used to detect the expression of LFCA in pPG-A3-LFCA/LRco21 (Lane 2) and culture supernatant (lane 3), and LRCON served as blank control (Lane 1) (see Appendix A). (**c**) Identification of LFCA content in culture supernatant of recombinant bacteria by ELISA. (**d**) Inhibition rate of LFCA expressed by the recombinant strain on the replication of SVA and FMDV. ** *p* < 0.01; * represents a significant difference between recombinant and blank strains.

**Figure 7 vetsci-12-00199-f007:**
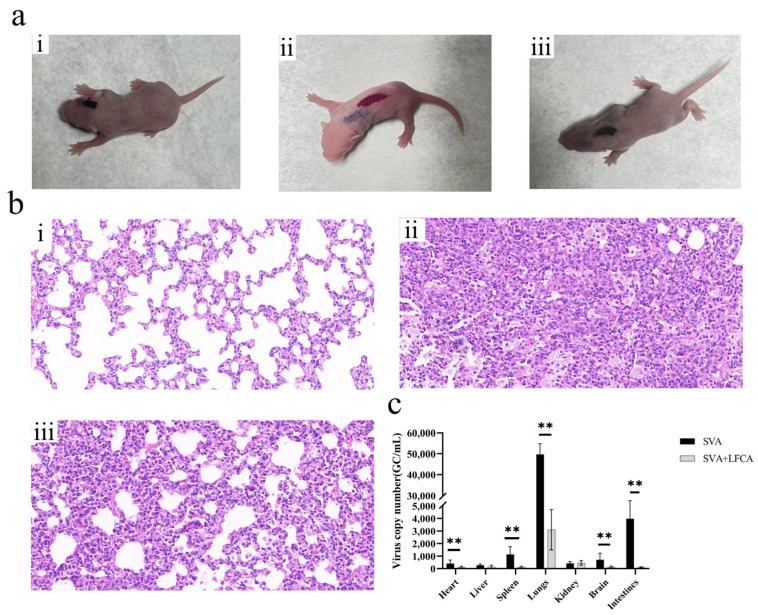
Pathological changes and viral loads in mice infected with SVA. (**a**) Symptoms of mice infected with SVA: (i) control; (ii) infection group; (iii) administration group. (**b**) Pathological changes in lung tissue: (i) control; (ii) infection group; (iii) administration group. (**c**) Viral load in different tissues. ** *p* < 0.01; * represents a significant difference between the dosing group and the infection group.

**Table 1 vetsci-12-00199-t001:** Pathological changes and viral load of mice infected with SVA.

Experimental Group	Earliest Onset of Symptoms (Days)	Number of First Symptoms (Only)	Mean Body Weight of Mice After Challenge (g)	Time to Symptom Relief (Days)	Number of Symptoms Relieved (Only)	Mean Body Weight of Mice at End of Experiment (g)	Changes in Body Weight of Mice (g)
CON	-	-	2.95	-	-	5.53	2.58 ± 0.114
SVA	1	7	2.97	4	7	5.23	2.26 ± 0.121
LFCA	2	8	2.83	4	8	5.30	2.47 ± 0.119 **

** *p* < 0.01; * significance of the difference between the representative and virus inoculation group.

## Data Availability

Data are contained within the article.

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
