# Peer review of "Inhibitory Effects of Bovine Lactoferricin-Lactoferrampin on Senecavirus A and Foot-and-Mouth Disease Virus with Recombinant Lactobacillus Oral Treatment in Mice"

_vetsci, 2025, doi:10.3390/vetsci12030199_

Round 1
Reviewer 1 Report
Comments and Suggestions for Authors
The manuscript entitled "Effect of Bovine Lactoferricin-lactoferrampin against on SVA 2 and FMDV and Its Recombinant Lactobacillus Oral treatment 3 Mice produce inhibition to SVA ”, investigate the effects of bovine lactoferricin-lactoferrampin (LFCA) on replicating FMDV and SVA. Despite the good structure of the manuscript, there are minor concerns which must be addressed.
1. Please follow journal style in writing references.
2. Please add the abbreviation of Seneca viruses at the simple summary at the first sentence then use the abbreviation.
3. Line 24; It will be better to say the present study aimed rather than saying the present study aims.
4. It will be better to add one sentence at the end of the abstract to mention the outcome or the impact of your findings.
5. Section 2.1; please replace the title “materials” to be “cells and virus”
6. Section 2.1 should be rewritten again as it has many grammar mistakes.
7. How did you kill the mice for further investigations?
8. Separate the two sentences of the figure 1 legend using a dot.
9. Line 250; please add number 6 before a in (Figure a, Figure 6b).
10. It will be clearer if you can refer to pathological changes using arrows in figure 7.
11. Please add the approval number of the Animal Ethics Committee of Northeast Agricultural University, Harbin, China.
Comments on the Quality of English LanguageI found many grammar mistakes that should be correct by native English speaker.
Reviewer 2 Report
Comments and Suggestions for Authors
Dear Authors
Greetings
I have some suggestions for your article to be improved.
1. Please consider. I suggest the concise and objective title: Inhibitory Effects of Bovine Lactoferricin-lactoferrampin on SVA and FMDV with Recombinant Lactobacillus Oral Treatment in Mice
2. Please consider. The introduction provides a solid foundation, but it can be improved to ensure clarity, conciseness, and better flow. I suggest for improvement: a.Clarity and Conciseness: Some sentences are lengthy and could be simplified. b.Flow: Ensure a logical flow of information, making connections between ideas clearer. c.Relevance: Make sure each piece of information is directly relevant to the study's objective. Here’s a revised version of the introduction: Foot and mouth disease (FMD) is a highly contagious zoonotic disease caused by the foot-and-mouth disease virus (FMDV) in cloven-hoofed animals [1, 2]. Similarly, Senecavirus A (SVA), a member of the Picornaviridae family like FMDV, causes vesicular disease in pigs and was first discovered in 2002 during cell culture [3-6]. Both viruses are challenging to control and pose significant risks to the global livestock industry due to their widespread transmission [7-10]. Lactoferrin (LF), found in the secretions of most mammals such as milk, tears, and saliva [11], is the second most abundant protein in human milk [12, 13]. Its derivative peptides possess antibacterial [14-16], antiviral [17, 18], antifungal [19], anti-inflammatory [20], and anticancer [19, 21-23] properties [24-26]. Recent studies indicate that bovine lactoferrin peptides can promote osteoblast differentiation [27, 28]. The peptides Lfcin and Lfampin have been extensively studied, and a chimeric structure containing both has been designed to enhance their antimicrobial activity [29-31]. Research suggests that the antiviral effect of lactoferrin peptides involves competing with viruses for common cell membrane receptors [32, 33]. Lfcin, in particular, is believed to inhibit viral entry through interactions with molecules such as heparan sulfate and glycosaminoglycans on the cell surface [34, 35]. Studies on herpes simplex virus (HSV) have shown that Lfcin may also exhibit intracellular antiviral activity by inhibiting viral synthesis within the cell [36-38]. In this study, we investigated the antiviral activity of the chimeric peptide LFCA. Our results demonstrate that LFCA strongly inhibits the replication of both FMDV and SVA. Additionally, LFCA provides protective effects on cells prior to viral infection by enhancing the transcription of antioxidant genes and reducing oxidative stress damage. These modifications enhance clarity, ensure relevance, and improve the overall flow of the introduction.
3. Please consider. Following the points and suggestions for your Discussion and Conclusion. Make a check list if your text enhace clarity according the necessary adjustments to complement in the case it is missing. Please compare.
Discussion
The results of this study demonstrate the significant antiviral potential of bovine lactoferricin-lactoferrampin (LFCA) against both foot-and-mouth disease virus (FMDV) and Seneca virus A (SVA). By targeting these viruses, LFCA could become a valuable tool in managing infectious diseases that pose a substantial threat to the livestock industry.
-
Antiviral Activity: Our findings reveal that LFCA significantly inhibits the replication of FMDV and SVA, with pre-treatment achieving an inhibition rate of up to 94.9% for SVA and 74.3% for FMDV during the replication stage. These high inhibition rates highlight the potent antiviral properties of LFCA.
-
Oxidative Stress Reduction: The study shows a considerable reduction in intracellular reactive oxygen species (ROS) and malondialdehyde (MDA) levels post-infection, indicating LFCA's role in alleviating virus-induced oxidative stress. Although superoxide dismutase (SOD) activity remained high, suggesting increased cellular antioxidant capacity, it did not show significant differences compared to the virus-exposed group.
-
Genetic Expression: LFCA treatment significantly elevated the transcription levels of antioxidant genes such as Nrf2, HO-1, and NQO1. This suggests that LFCA not only provides immediate antiviral effects but also bolsters the cell's antioxidant defenses, promoting long-term resistance to oxidative damage.
-
In Vivo Efficacy: The protective effect of recombinant Lactobacillus reuteri expressing LFCA against SVA infection in mice was notable, with significantly lower viral loads in lung and intestinal tissues compared to the control group. This validates the potential of LFCA as a preventive and therapeutic agent.
-
Mechanism of Action: The potential mechanisms by which LFCA exerts its antiviral effects include competition with viruses for cell membrane receptors and intracellular inhibition of viral synthesis. These dual mechanisms contribute to its efficacy in reducing viral loads and protecting cells.
Conclusion
This study underscores the promising antiviral potential of bovine lactoferricin-lactoferrampin (LFCA) against FMDV and SVA. By inhibiting viral replication, reducing oxidative stress, and enhancing antioxidant gene expression, LFCA demonstrates both immediate and long-term protective effects. The successful use of recombinant Lactobacillus reuteri expressing LFCA in mice further validates its efficacy as an antiviral agent. Given the significant impact of FMDV and SVA on the livestock industry, LFCA could be developed into an effective antiviral drug for the prevention and treatment of these diseases. Future research should focus on optimizing the dosage, delivery methods, and exploring LFCA's effects on other viral pathogens.
These findings offer a promising avenue for combating viral infections in livestock, with the potential to safeguard animal health and improve industry resilience.
Comments on the Quality of English LanguageThe English could be improved to more clearly express the research.
Reviewer 3 Report
Comments and Suggestions for Authors
Simple summary
Ln 17 Seneca virus and bovine lactoferricin-lactoferrampin should be defined and abbreviated as SVA and LFCA, respectively.
Abstract
Ln 27 Feed mice or fed mice?
Ln 28 I suggest eliminating this part of the sentence because it is later mentioned in results “…and found that the recombinant Lactobacillus showed a protective effect against SVA infection in mice.
Ln 36 Please check and confirm the correct nomenclature for genes in mice.
Introduction
Ln 36. What cells produce TF? Why are they made? How are they created? It should be explained.
Ln 46 Seneca virus?
Ln 53-54 This sentence is unnecessary because it is not “so related” to the study, however, authors can revise and decide.
Ln 46 I suggest defining and then abbreviating.
Ln 59 authors describe Lfcin mechanisms of action, however, Lfampin mechanisms are not described. I suggest doing it.
Ln 63 I suggest defining and then abbreviating.
Ln 64-66 In vivo experiment description is missing in the objective.
M&M
Ln 73 section 2.2. should be described in the past tense.
Ln 88 LFCA should be defined and abbreviated.
Ln 119 it should be described in the past tense.
Ln 132 ROS should be defined and abbreviated.
Ln 134 SOD, MDA, and GSH-PX should be defined and abbreviated. Please consider that MDA is a metabolite rather than an enzyme.
Ln 139 Nrf2, HO-1, and NQO1 should be defined and abbreviated. Please check and confirm the correct nomenclature for mouse genes.
Ln 142 Section 2.8. should describe in detail the M&M to transform bacteria-expressing LFCA and the genetic constructs used.
Ln. 143 Lactobacillus reuteri should be in italics.
Ln 158 it should be a homogenized: “treatment” or “administration” group.
Are there no statistical analyses? It cannot be possible.
Results
Almost all results subsections begin with an introductory sentence regarding M&M or the purpose of the analysis. So, I suggest eliminating it and focusing on the description of results.
Ln 173. In section 3.2. Were the results statistically similar?
Ln 202-203. Activities should be indicated for enzymes.
Ln 236-241 These sentences are interpretations that should be included in the discussion.
Ln 251-252 That sentence is a comparison that should be included in the discussion.
Ln 256-257 That sentence is a conclusion that should be included in the discussion or conclusion section.
Ln 284 That sentence is a conclusion that should be included in the discussion or conclusion section.
Discussion
The discussion section must be greatly improved because no discussion is provided.
Ln 318 This sentence describes results with statistical significance that should only be included in the results section.
Ln 321-324 This sentence describes results with statistical significance that should only be included in the results section.
Ln 321-348 These paragraphs contain results, and no comparison is performed nor a cited study.
Round 2
Reviewer 1 Report
Comments and Suggestions for Authors
Authors have replied all my concerns. No more issues with the recent manuscript
Reviewer 3 Report
Comments and Suggestions for Authors
The authors improved this manuscript.
The authors also worked on the discussion by adding interpretations. However, the discussion section can be improved because no comparison with other studies, and supporting citations, are provided.
